# Electrochemical (Bio)Sensors for Pesticides Detection Using Screen-Printed Electrodes

**DOI:** 10.3390/bios10040032

**Published:** 2020-04-02

**Authors:** Beatriz Pérez-Fernández, Agustín Costa-García, Alfredo de la Escosura- Muñiz

**Affiliations:** NanoBioAnalysis Group-Department of Physical and Analytical Chemistry, University of Oviedo, Julián Clavería 8, 33006 Oviedo, Spain

**Keywords:** screen-printed electrodes, electrochemical (bio)sensors, pesticides, point-of-care, food control

## Abstract

Pesticides are among the most important contaminants in food, leading to important global health problems. While conventional techniques such as high-performance liquid chromatography (HPLC) and mass spectrometry (MS) have traditionally been utilized for the detection of such food contaminants, they are relatively expensive, time-consuming and labor intensive, limiting their use for point-of-care (POC) applications. Electrochemical (bio)sensors are emerging devices meeting such expectations, since they represent reliable, simple, cheap, portable, selective and easy to use analytical tools that can be used outside the laboratories by non-specialized personnel. Screen-printed electrodes (SPEs) stand out from the variety of transducers used in electrochemical (bio)sensing because of their small size, high integration, low cost and ability to measure in few microliters of sample. In this context, in this review article, we summarize and discuss about the use of SPEs as analytical tools in the development of (bio)sensors for pesticides of interest for food control. Finally, aspects related to the analytical performance of the developed (bio)sensors together with prospects for future improvements are discussed.

## 1. Introduction

Pesticides are among the most used products in the agri-food industry for the control, prevention and elimination of pests. According to the target pest, they can be classified in insecticides, acaricides, fungicides, bactericides, herbicides, etc. The main pesticides are made of carbamates, dinitrocompounds, organochlorines, organophosphates, pyrethroids, neonicotinoids or triazines, among others [1]. However, such compounds have a high toxicity. In this line, according to the World Health Organization (WHO), they can be classified as carcinogenic, neurotoxic or teratogenic [2,3]. This makes necessary their strict control in wastewater, soil, food, animals and human beings. In the European Union, the Residual Maximum Limits (MRLs) allowed by the legislation are 0.1 µg/L for individual pesticides and 0.5 µg/L for total pesticides [4,5,6]. 

The United Nations (UN) estimates that 200,000 deaths from acute poisoning occur each year due to pesticides, 99% belonging to developing countries [7]. Continuous exposure to these compounds may cause cancer, Alzheimer’s disease and Parkinson’s disease, as well as neurological disorders, fertility issues, allergies and hypersensitivity.

The official methods for the determination of pesticides are based on chromatography, such as High Performance Liquid Chromatography (HPLC), HPLC-MS/MS and Gas Chromatography coupled to Mass Spectrometry (GC-MS/MS) [8,9,10,11,12,13]. Despite their high sensitivity, these techniques require very expensive equipment, long analysis times, high reagent sample volumes and qualified personnel. Due to these limitations, alternative methodologies for pesticides detection have been proposed in the last years, being probably the most relevant ones those based on electrochemical methods [14]. Electrochemical techniques have advantages over conventional ones related to their simplicity, the low sample volumes required (typically in the order of µL), the low cost of instruments and the short analysis time [15,16]. The main (bio)sensing routes for the electrochemical detection of pesticides are based on (Figure 1): (i) enzymatic sensors (inhibition and enzymatic catalysis); (ii) direct detection-based sensors (of electroactive pesticides); (iii) immunosensors (using specific antibodies as receptors); iv) aptasensors (using specific aptamers as receptors) and (v) biological sensors (using microorganisms as receptors). 

However, the use of traditional electrodes requires relatively high sample volumes and quite complicated cell set-up, as they are not suitable for “in field” applications. In this sense, screen-printed electrodes (SPEs) have emerged as outstanding alternatives, overcoming the limitations of traditional electrodes. Screen printing is a well-developed technology widely used since the 1990s for the mass production of disposable and economical electrochemical sensors. This production process is carried out in several stages, as illustrated in Figure 2A [17]. 

Screen-printed electrodes (SPEs) are manufactured on ceramic or plastic substrates, in which different types of inks (typically carbon, graphite, silver and gold) are printed. In addition, these inks can be modified with nanomaterials or enzymes among other compounds, improving the analytical characteristics of the (bio)sensors developed from such electrodes.

SPEs satisfy the need for highly reproducible, sensitive and cost-effective detection methods, with additional advantages related to the low cost of production, flexibility in design, small size and ease of electrode surface modification [18,19,20,21,22,23]. The main methods used for the immobilization of (bio)receptors on the working electrode of SPEs for further pesticides detection are summarized in Figure 2B. The portability of the electrochemical instruments typically used also makes these systems ideal for point of care (POC) analysis [24,25,26,27]. The working electrode can be modified with various materials and recognition elements such as noble metal nanoparticles (i.e., Cu, Ni, Au, Pt, Ag) [28,29,30,31,32,33,34,35,36,37], nanotubes (CNT) [38,39,40,41,42,43,44], nanofibers (CNF) [43,45,46,47,48,49], graphene [50], graphene oxide (GO) [51,52,53,54], reduced graphene oxide (rGO) [55,56,57,58], quantum dots (QDs) [59,60,61,62,63,64], magnetic beads (MB) [65,66,67,68], enzymes (AChE, ALP, GOD, HRP, FDH, OPH, Tyr) [69,70,71,72,73,74,75,76,77,78,79,80], antibodies [81,82,83,84,85,86,87], aptamers [88,89,90,91,92,93], DNA [94,95,96,97,98] and biological agents [99,100,101].

In this review, recent applications of SPEs for the electrochemical detection of pesticides are summarized, giving a critical vision on the advantages, drawbacks and perspectives.

## 2. Electrochemical Techniques Used for Pesticides Detection 

The typical electrochemical techniques used for pesticides detection, after following one of the (bio)sensing routes schematized in Figure 1, are: voltammetry (cyclic voltammetry, differential pulse voltammetry, square wave voltammetry), chronoamperometry and electrochemical impedance spectroscopy. The main characteristics of each technique are briefly described in this section. 

### 2.1. Cyclic Voltammetry

Cyclic voltammetry (CV) is used to study the different electrochemical processes that take place when applying a potential scan. The measured peaks of current provide information on the oxidation and reduction processes of an electroactive specie. In addition, it provides information on the type of process object of study: (i) reversible; (ii) irreversible or (iii) quasi-reversible, depending on the separation between the anodic and cathodic peaks. Another characteristic that popularizes the CV is its ability to give information about the nature of a process in terms of adsorption and diffusion characteristics.

### 2.2. Differential Pulse Voltammetry 

Differential pulse voltammetry (DPV) technique consists in applying a sequence of pulses of constant amplitude superimposed on a stepped potential increase. The current intensity is measured just before applying the pulse and at the end of it. The response obtained is the difference between the two current intensities, in relation to the potential at the start of the pulse, giving rise to a peak-like response.

This technique is used to determine oxidation or reduction processes depending on the analyte concentration. In general terms, DPV has better sensitivity than cyclic voltammetry.

### 2.3. Square Wave Voltammetry

In square wave voltammetry (SWV) technique, a large amplitude square wave potential sweep is applied with a stepped potential ramp. Current intensity is measured at the end of each applied pulse in the potential sweep cycle.

Generally, SWV is more sensitive, faster and more selective than DPV, since the background current is minimized.

### 2.4. Chronoamperometry

Chronoamperometry (AC) is a determination technique where a constant current intensity is applied for a certain time. During to the application of such potential, the electroactive analytes present in the solution are oxidized or reduced, generating and associated current, proportional to the amount of analyte. The main advantage of this technique compared with voltammetries is related to its simplicity.

### 2.5. Electrochemical Impedance Spectroscopy

Electrochemical impedance spectroscopy (EIS) is a technique commonly used to evaluate parameters of charge transfer, corrosion processes, double-layer formation, or modification processes of the electrode surface.

The EIS is given by the Nyquist diagram where values of the load transfer resistance and the resistance of the solution are obtained. Depending on the semicircle obtained in the Nyquist diagram, it can be determined if there is an impediment to the charge transfer or if it is favored by the modification of the working electrode. This technique is widely used in label-free biosensing formats, where the change in the impedance upon the analyte biorecognition allows its determination.

## 3. Enzymatic Sensors

Enzymatic sensors are the most widely used for the determination of pesticides on SPEs. Two are the main detection routes based on enzymes: (i) enzymatic inhibition route, where the pesticide inactivates the enzyme and (ii) catalytic route, where the pesticide is hydrolyzed by the enzyme generating an electroactive compound [102,103]. 

### 3.1. Enzymatic Inhibition

As stated above, inhibition reactions make enzymes inactive in the presence of pesticides. This is the case of acetylcholinesterase (AChE), butyrylcholinesterase (BChE), tyrosinase (Tyr) and alkaline phosphatase (ALP).
(1)$$\mathrm{Acetylcholine}+{\;H}_{2}O\;\overset{\;\mathrm{AChE}\;}{\to }\mathrm{Choline}+\mathrm{Acetate}$$

AChE is one of the most commonly used enzymes for the determination of organophosphorus pesticides (OP). The reaction catalyzed by AChE is the following:

The presence of OP results in the inhibition of the enzymatic activity due to phosphorylation of the serine residue of the active center of the enzyme, which blocks the hydrolysis of acetylcholine (ACh). Therefore, the higher the concentration of OP, the more blocked the active center will be and the lower the signal of oxidation or reduction of the enzymatic products. Detection based on other enzymes relies in the same principles.

As shown in Table 1, many publications on enzymatic inhibition-based detection of pesticides on SPEs have been reported in the last years. For example, Solna et al. [74] developed a multi-analyte device for pesticides and phenols based on the immobilization of enzymes (AChE, BChE, Tyr, HRP) on different graphite and platinum working electrodes. AChE and BChe were immobilized on platinum working electrodes while Tyr and HRP did on graphite working electrodes. The limits of detection (LoDs) for the different pesticides studied depended on whether AChE or BChE were used, ranging from 0.8 to 130 nM for AChE and 2.8 to 2390 nM for BChE. Such difference in LoDs is related to the different affinity of each enzyme for the corresponding pesticide. Industrial wastewater was successfully analyzed with such system. Another biosensor based on the inhibition of AChE on screen-printed carbon electrodes (SPCEs) was the developed by Dou et al. [104], where after the manufacturing of the SPCEs, the enzyme was immobilized by polyacrylamide polymerization. Dichlorvos, Monocrotophs and Parathion pesticides were determined with LoD of 18.1, 26.4 and 14.4 nM respectively. In both works the enzymes were immobilized on unmodified working electrodes, giving rise to simple and fast sensors.

However, in most cases working electrodes are modified with different materials so as to improve the efficiency of the enzyme immobilization and thus the pesticide analysis. An example of simple electrode modification is the reported by Arduini et al. [105] where Prussian Blue was immobilized before the enzyme (AChE and BChE) did. The concentration of several organophosphorus pesticides was determined, finding the highest sensitivity for Aldicarb and Carbaryl (LoDs 63 and 124 nM respectively) when AChE was used and for Paraoxon and Chlorpyrifos-methyl oxon (LoDs 7 and 1.6 nM, respectively) when using BChE. River water and wastewater samples were analyzed with such biosensor. Polymers like poly (3,4-ethylenedioxythiophene) polycation (PEDOT) and poly (styrenesulfonate) polyanion (PSS) were also used for the carbon electrode modification so as to increase its conductivity [106]. Organophosphorus pesticides such as Chlorpyrifos-oxon were determined (LoD of 4.4 nM) based on the thiocholine oxidation. In the case of the biosensor developed by Silva Nunes et al. [107], a screen-printed graphite electrode (SPGE) was modified with 7,7,8,8-tetracyanoquinodimethane (TCNQ) and photopolymerized with poly (vinyl alcohol) bearing styrylpyridinium groups (PVA-SbQ) to covalently immobilize AChE. This biosensor was applied for the determination of different carbamates such as Aldicarb (LoD 8 nM), Carbaryl (LoD 4 nM), Carbofuran (LoD 1 nM) and Methomyl (LoD 2 nM).

Different nanomaterials have also been proposed for the electrode’s modification. For example, multi-walled carbon nanotubes (MWCNTs) were immobilized together with AChE and Co-phthalocyanine (Co-Pc; used as mediator) on the working electrode of a SPCE [108]. Such modification produces a decrease in the working potential, minimizing interferences and thus improving the selectivity of the biosensor. This device was developed for the determination of Paraoxon (LoD 11 nM) and Malaoxon (LoD 6 nM), even in tap and sparkling water. Metallic nanoparticles (NPs) have also been used as modifiers of the working electrode. This is the case of the Clortoluron sensing device developed by Haddaoui and Raouafi [109] using ZnO nNPs-modified electrodes for the immobilization of tyrosinase (Tyr) enzyme (Figure 3). Clortoluron is detected here with a LoD of 0.47 nM, having also a good performance in tap water, well water and river water. The stability of immobilized AChE as well as the electronic transference were also improved using magnetic nanoparticles (Fe_3_O_4_) coupled to a graphene (GR) film on a SPCE [110]. This biosensor was applied for the determination of Chlorpyrifos at levels of 0.06 nM even in vegetable samples (spinach and cabbage).

Alternatively, enzymes can also be trapped in gel matrices with which the electrode is subsequently modified. This is the case of the work developed by Shi et al. [111] where AChE is trapped in a sol-gel matrix of Al_2_O_3_. Such matrix not only increases the stability of AChE but also catalyzes the oxidation of thiocholine, decreasing the working potential and minimizing interferences. Dichlorvos pesticide was determined at levels of 0.8 µM in river water samples.

As summarized in Table 1, the modification of SPEs with different materials highly improves the analytical characteristics of the biosensors, lowering at levels as low as the femtomolar scale.

### 3.2. Catalytic Detection

As stated above, an alternative route for pesticides detection using enzyme receptors is based on the pesticide hydrolysis by the enzyme generating an easily detected electroactive compound. Organophosphorus anhydrolase acid (OPAA) and organophosphorus hydrolase (OPH) are the main enzymes used for such purpose [143,144,145,146,147]. The OPAA is a more restrictive enzyme with respect to organophosphorus compounds, being only able of catalyzing P-F bonds, while OPH also catalyzes organophosphorus containing P-O, P-S and P-CN bonds [148]. Therefore, OPH has a broader range of catalytic effect, being used to develop biosensors for total organophosphorus pesticides.

The catalytic reaction of OP through the use of OPH gives rise to the electroactive p-nitrophenol product. Consequently, the amount of OP is proportional to the production of p-nitrophenol. The generic reaction catalyzed by OPH is as follows: (2)$$\mathrm{Aryldialkyl}\;\mathrm{phosphate}+{\;H}_{2}O\;\overset{\;\mathrm{OPH}\;}{\to }\mathrm{Dialkyl}\;\mathrm{phospahte}+\mathrm{Aryl}\;\mathrm{alcohol}$$

Table 2 summarizes the different biosensors reported for organophosphates detection based on enzymatic hydrolysis. As in the case of the enzymatic inhibition, different materials have been evaluated for the electrode modification so as to improve the performance of the biosensor.

As a representative example, Mulchandani et al. [149] modified the SPCE with Nafion (Nf) for the OPH immobilization and subsequent determination of Paraoxon and Methyl parathion in river water samples at 0.9 and 0.4 µM levels respectively. They and others [150], also evaluated the SPCE modification with MWCNT for the enzyme immobilization finding worse sensitivity than for the Nf-based modification.

Metallic NPs have also been proposed for the electrode modification. In particular, AuNPs surrounding the core of magnetic Fe_3_O_4_ NPs were used for the immobilization of methyl parathion hydrolase enzyme (MPH) which allowed the detection of Methyl parathion at levels as low as 0.38 nM (Figure 4) [151]. The use of the AuNP/ Fe_3_O_4_ NPs matrix also allowed to work at low potentials, minimizing interferences when analyzing river water samples.

## 4. Direct Detection of Electroactive Pesticides

In spite of their advantages, the use of enzymes entails important limitations related to their low stability over time and their sensitivity to changes in temperature and pH [154]. Moreover, some pesticides are electroactive compounds able to be electrochemically detected without the need of enzymes [155,156]. For this reason, the growth of electrochemical sensors without enzymes is continuously increasing, benefiting of their lower cost, greater simplicity and faster analysis. However, in most cases the sensitivity of such direct detection on unmodified SPEs is not enough for detecting the pesticides at the maximum levels allowed by the legislation. Working electrode modification with different materials, mainly metallic nanoparticles (made of i.e., Ag, Zn, Cu, Ni) and carbon nanomaterials such as nanofibers (CNF), nanotubes (CNT) or graphene, etc. has been extensively studied so as to improve the electronic transference and thus the sensitivity of the pesticide detection. 

Direct detection approaches for pesticides determination on SPEs are summarized in Table 3. As a representative example of direct detection without electrode modification, Geto et al. [157] self-prepared carbon made SPEs (SPCE) (Figure 5A) for the rapid determination of Bentazone (BTZN), an important herbicide used in agriculture. The voltammetric oxidation of the tertiary amine in the pesticide was selected as the analytical signal that allowed its quantification. A detection limit below the MRL (3.4 nM) was obtained, also demonstrating the good performance of the sensor in groundwater and lake water.

However, as stated above, in most cases SPE modification is required for improving the sensor sensitivity. In this line, Della Pelle et al. [158] used black nanocarbon (CB) to develop a device for the determination of phenylcarbamates (i.e., Carbaryl, Carbofuran, Isoprocarb and Phenobucarb). In this case, the analytical signal corresponds to the voltammetric oxidation of the hydrolyzed forms of the pesticides, which allowed to reach LoD ranging from 48 to 80 nM. Wheat and corn samples were also analyzed after extraction and hydrolysis treatment.

Thick bismuth films have also been proposed as SPEs modifiers for the determination of neonicotinoid pesticides such as Clothianidin, Imidacloprid, Thiamethoxam and Nitenpyram [159]. The voltammetric reduction of the nitro group of the pesticides to hydroxylamine allowed the use of quantitative analysis, reaching LoDs at levels in the range 2.97–4.12 μM. Tap water, mineral water and samples from rivers and lakes were successfully analyzed. 

NiO NPs have also been used to improve the sensitivity and stability for the determination of Parathion (Figure 5B) [160]. Such NPs catalyze the voltammetric reduction of the nitro group of the Parathion to hydroxylamine, allowing the detection of the pesticide at 24 nM levels. Tap water and human urine samples were analyzed without any pretreatment while a tomato juice sample only required a simple filtration.

AuNPs were also used to increase the electrode surface of the SPCE and catalyze the amperometric oxidation of Thiram, Disulfiram and N,N-diethyl-N’,N’-dimethylthiuram disulfide (DEDMTS) pesticides [161]. Interestingly, in this case, this sensor was coupled to an Ultra High-Performance Liquid Chromatography (UHPLC) system to perform a previous separation and improve the selectivity. LoDs ranging from 0.05 to 0.55 µM were obtained with such system, also analyzing samples of apple, grape and lettuce after extraction, filtration and centrifugation.

The high selectivity given by the use of molecularly imprinted polymeric membranes (MIP) has also been approached for the specific pesticide detection. As an example, a MIP combined with AuNPs and reduced graphene oxide (rGO), was used for the specific voltammetric determination of Cyhexatin (CYT) [162] (Figure 5C), reaching a LoD as low as 0.52 nM. 

The SPEs electrodes used by Jubete et al. [163] were modified with single-wall carbon nanotubes (SWCNT) and cobalt phthalocyanine (CoPc) for the determination of Thiocholine (TCh), by monitoring its amperometric oxidation. This system gave a relatively low sensitivity with LoDs at 38 µM levels. In contrast, when zinc oxide nanoparticles (ZnONP) were used in combination with multi-walled carbon nanotubes (MWCNTs) [164] for the determination of Glyphosate and its hydrolysis product (aminomethylphosphonic acid, AMPA) the LoDs were significantly better (300 nM and 3 µM).

## 5. Immunosensors

Immunosensors for pesticides detection are based on the use of antibodies and antigens as recognition elements immobilized on the SPE. The main advantages of such biosensors over enzymatic ones rely on the higher stability of antibodies/antigens together with greater selectivity and specificity. However, the high cost and low availability of monoclonal antibodies is an important limitation that should be considered. The small size of the pesticides also usually avoids sandwich-based approaches, so competitive immunoassays are often required. Immunosensors for Imidacloprid, Parathion, Methyl Chlorpyrifos, Chlorsulfuron and Atrazine can be found in the literature, as shown in Table 4. 

In most cases, SPEs are modified with different (nano)materials so as to improve the efficiency of the receptor immobilization. As example, graphene sheets [176] and graphene quantum dots (GQDs) [177] modified with amino groups have been proposed for the oriented immobilization of antibodies on SPCE. In both cases, anti-parathion antibodies were immobilized for the further Parathion recognition and final detection by means of electrochemical impedance spectroscopy (EIS), reaching LoDs as low as 0.16 pM, with high selectivity even in tomato and carrot samples after extraction (Figure 6A).

Representative examples of competitive immunosensors are those based on the use of enzymatic tags proposed for the determination of Chlorsulfuron [178] and Imidacloprid [179] at levels as low as 22 pM, also taking advantage of the use of AuNPs [180] for the immobilization of the antibody. Tap water, watermelon and tomato samples were analyzed here without the need of pretreatment (Figure 6B).

## 6. Aptasensors

Aptamers are single-chain oligonucleotides that can be produced through the technique of “systematic evolution of ligands by exponential enrichment” (SELEX) [184]. These aptamers are able to fold into three-dimensional structures to bind small compounds such as pesticides and drugs or large organisms [185,186,187,188,189,190]. Aptamers and analytes are joined by Van der Waals forces, electrostatic interactions or hydrogen bonds [191], thus being able to reverse the aptamer/analyte bond. Some of the advantages of the use of aptamers are the lower cost with respect to enzymes and antibodies, higher stability, long service life, regeneration possibilities and simplicity and rapid response [185,192]. Aptamers can also be easily functionalized and immobilized on the SPE for the development of electrochemical aptasensors.

However, their use for the detection of pesticides has not been extensively reported, as shown in Table 5. Again, SPE modification with different (nano)materials seems to be crucial for improving the efficiency of both the aptamer immobilization and the electrochemical detection.

As example, polyaniline/AuNPs composites were proposed by Rapini et al. [193] for the immobilization of an acetamiprid-specific aptamer (Figure 7A). The pesticide detection was based on a competitive assay using an enzyme-tagged oligonucleotide complementary to the aptamer sequence, reaching a LOD of 86 nM with high selectivity. Good performance was also obtained when analyzing blackberry, apricot and peach juice samples.

Aptamers have also been immobilized on SPE in composites with reduced graphene oxide (rGO) and Cu nanoparticles (CuNP). That is the case of the work reported by Fu et al. [194] for the determination of organophosphorus pesticides. The voltammetric signal decreases when increasing the pesticide concentration, since the complex aptamer-pesticide hindered the transfer electron of the [Fe(CN)_6_]^3-/4-^ ions. Under the optimal conditions LoDs ranging from of 0.003 to 0.3 nM were obtained for Profenofos, Phorate, Isocarbophos and Omethoate. The aptasensor was also successfully applied for rapeseed and spinach samples, after an extraction treatment.

AuNP-modified SPEs were also proposed for the immobilization of an aptamer specific for Diazinon (DZN) [195]. Increase in the impedance upon pesticide recognition was approached for its determination at levels as low as 17 fM, also taking advantage of the AuNPs as enhancers of the electronic transference (Figure 7B). Rat plasma samples were also analyzed with good sensor performance. 

## 7. Biological Sensors

Cells and microorganisms can also be used as recognition elements in biosensors. In this case the analytical signal is commonly related to the activation or inactivation of cellular respiration upon analyte interaction, leading to the production of electroactive metabolites. A representative approach consists in the genetic modification of microorganisms with enzymes, such as OPH for the determination of organophosphorus pesticides by measuring the enzymatically produced p-nitrophenol [102,196,197,198].

However, a key limitation of using microorganisms as a recognition element relies in their low sensitivity and long analysis time because of the slow transport of substrate and product through the cytoplasmic membranes of the cells. Due to this, few works on the use of microorganisms for the development of pesticide biosensors on SPE are found in the literature, as summarized in Table 6. From these works, the report by Touloupakis et al. deserves to be highlighted [199] which employed Photosystem II (PS II) that has an oxidoreductase-like behavior. In this work, they used the photosynthetic thylakoid of *Spinacia oleracea*, *Senecio vulgaris* and its atrazine resistant mutant immobilized with BSA-GA on the SPE for the detection of herbicides that selectively block the electronic transfer activity of PS II biomediators (Figure 8A). The developed multi-biosensor reached LoDs at levels ranging from 15 to 41 nM for Diuron, Atrazine, Simazine, Terbuthylazine and Deethylterbuthylazine, even in river water samples.

Alternatively Chatzipetrou et al. [200] used as recognition elements the bacterial reaction centers (RC) of *Rhodobacter sphaeroides* immobilized on gold SPE for the determination of Terbutryn at 8 nM levels (Figure 8B). 

Finally, the biosensor reported by Kumar and D’Souza [201] is based on the immobilization of whole cells of recombinant *Escherichia coli* on an SPCE for the detection of Methyl parathion. The organophosphorus hydrolase enzyme that catalyzes the hydrolysis of organosphorus pesticides such as Methyl parathion in p-nitrophenol is expressed in recombinant *Escherichia coli* cells, being an electroactive compound through which the concentration of Methyl parathion is directly determined. With this approach, Methyl parathion was detected at 0.5 µM levels. 

## 8. Conclusions

Screen-printed electrodes (SPEs) are emerging platforms with outstanding potential for their use as transducers in electrochemical (bio)sensing of pesticides. Their well-known advantages in terms of disposability, portability and low-volumes required, among others, make them ideal for the “in field” detection of pesticides at the point of need. Moreover, their versatility and easy modification with different materials is of key relevance for reaching ultralow detection limits that allow to detect the pesticides at the maximum levels allowed by the legislation. In this line, the use of nanomaterials, such as carbon-related ones (graphene, carbon nanotubes) and metallic nanoparticles is the object of an extensive research in recent years.

The (bio)sensing strategy to be followed for the pesticide detection must be carefully studied for each concrete case.

The simpler and faster strategy consists in the direct detection, taking advantage of the electroactivity of some pesticides, that is the presence of functional groups with red-ox properties. However, a limited group of pesticides can be sensitively detected through this route, typically at levels of µM–nM. 

Enzymatic sensors are the most widely used, benefitting from the wide range of pesticides able to be detected and the high sensitivity reached, typically at levels of nM–pM. However, the use of enzymes entails important limitations related to their low stability over time and their sensitivity to changes in temperature and pH, among others, so alternative biosensing methods based on antibody receptors, are becoming popular in the last years.

The main advantages of such immunosensors over the enzyme-based ones rely on the higher stability of antibodies together with their superior selectivity and specificity. The detection limits reached are quite similar to the ones obtained through the enzymatic route, typically at pM levels. The high cost and low availability of monoclonal antibodies and the small size of the pesticides that usually avoids sandwich-based approaches are important limitations that should be considered before selecting this biosensing approach. 

In conclusion, the selectivity and sensitivity/detection limit required, and the availability of specific enzymes and antibodies are the main parameters that should define the detection strategy more suitable for each particular application.

As far as we know, none of the reported electrochemical (bio)sensors for pesticides detection on SPEs are commercially available yet. In this line, some important issues should be solved for the implantation of such (bio)sensing systems for routine analysis, as alternative to centralized laboratory-based methods (HPLC-MS/MS; GC-MS/MS). Efficiency and long-term stability of the enzymes and the antibodies are crucial issues that are not addressed in most of the reviewed works. Moreover, multi-detection abilities should be strongly required for real applications in pesticides screening. Efforts in this sense should be the next at the current state of the art.

Overall, the higher potentiality of SPE-based pesticide (bio)sensors is in the decentralized “in field” analysis, in our opinion. The combination of such miniaturized electrochemical transducers, the cheap and portable electrochemical instruments and the stability of mainly the antibodies make altogether ideal for such applications. In the case of the immunosensors, the most challenging issue is related to the sampling, washing, etc. steps required, which limits their use by non-skilled people and consequently their commercial implantation. The combination with microfluidics seems to be of key relevance here, so high efforts in this sense are previewed for the coming years. 

## Figures and Tables

**Figure 1 biosensors-10-00032-f001:**
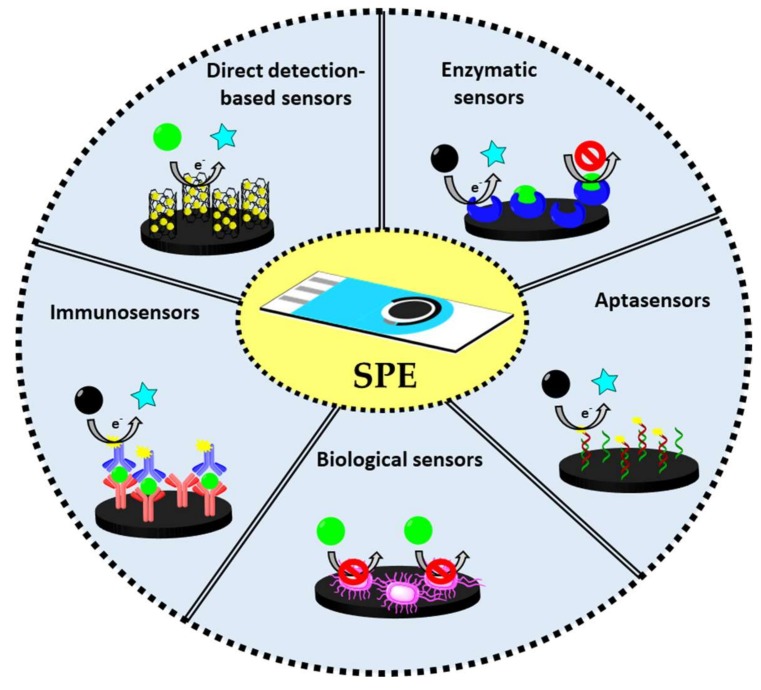
Main (bio)sensing routes followed for the electrochemical detection of pesticides.

**Figure 2 biosensors-10-00032-f002:**
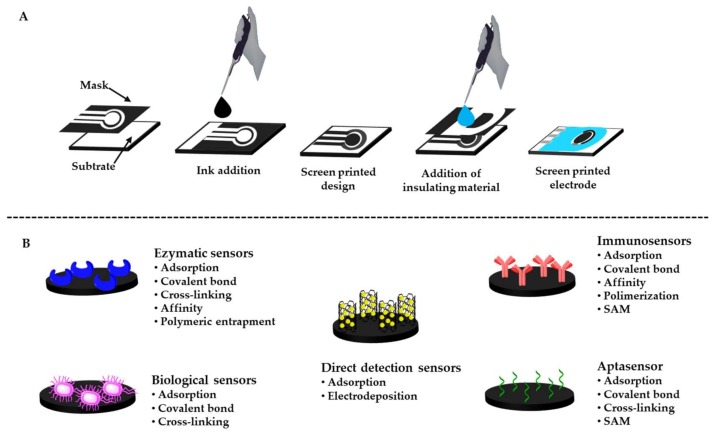
(**A**) Stages of the manufacturing process of screen-printed electrodes (SPEs). (**B**) Main methods used for the immobilization of (bio)receptors on the working electrode of SPEs for pesticides detection (SAM: Self Assembled Monolayers).

**Figure 3 biosensors-10-00032-f003:**
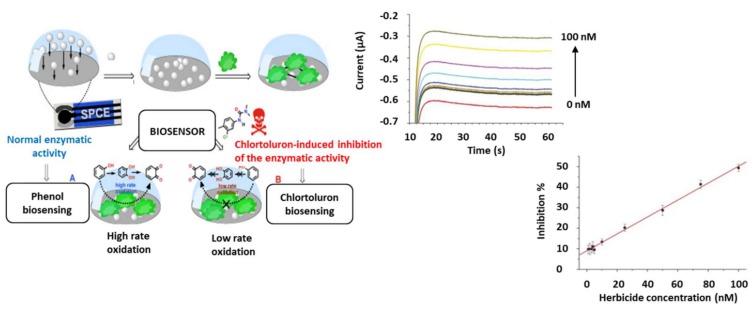
Enzyme inhibition biosensors for pesticides detection on screen-printed electrodes. (Left) scheme of a biosensor for Chlortoluron using a SPCE modified with ZnO NPs for Tyr enzyme immobilization; (Right) Chronoamperometric (CA) responses and calibration curve of inhibition % vs. concentration of herbicide. Reprinted from [109], Copyright 2015, with permission from Elsevier.

**Figure 4 biosensors-10-00032-f004:**
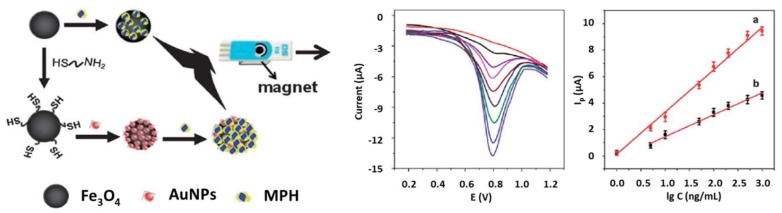
Enzymatic hydrolysis-based biosensors for pesticides detection on screen-printed electrodes. (Left) Scheme of an approach based on the SPCE modification with AuNPs/ Fe_3_O_4_ NPs for MPH immobilization and further determination of methyl parathion; (Right) SWV measurements of methyl parathion at different concentrations and calibration plots for the electrode with (**a**) and without (**b**) AuNPs. Reprinted from [151], Copyright 2013, with permission from Royal Society of Chemistry.

**Figure 5 biosensors-10-00032-f005:**
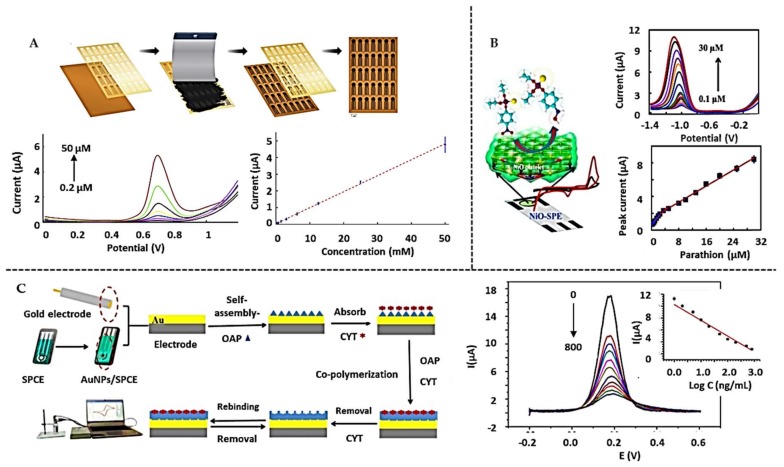
Direct detection-based biosensors for pesticides determination on screen-printed electrodes: (**A**) Scheme of the preparation of SPE for the determination of BTZN, SWV measurements and calibration curve. Reprinted from [157], Copyright 2019, with permission from Elsevier. (**B**) SPE modified with NiO NPs to determine Parathion, DPV signals and calibration curve. Reprinted from [160], Copyright 2018, with permission from Elsevier. (**C**) Use of MIP for the determination of CYT. Measurements by DPV and inset the calibration curve. Reprinted from [162], Copyright 2019, with permission from Elsevier.

**Figure 6 biosensors-10-00032-f006:**
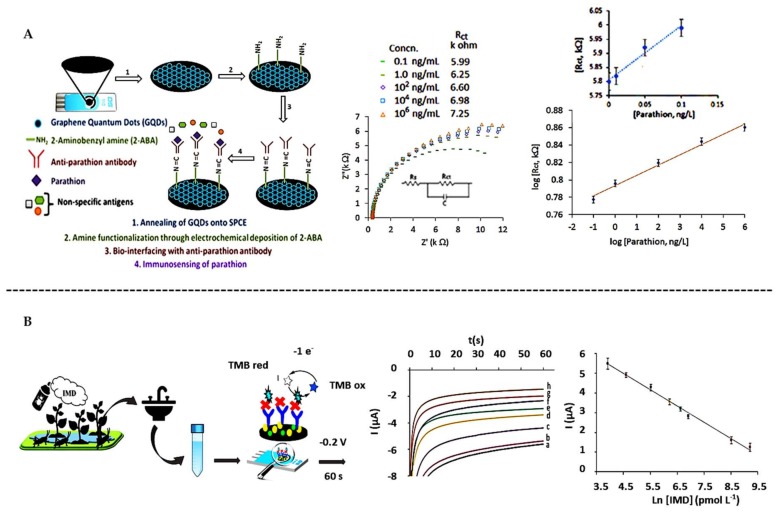
Immunosensors for the determination of pesticides on screen-printed electrodes. (**A**) Use of GQDs to determine Parathion, Niquist curves and calibration curve. Reprinted from [177], Copyright 2017, with permission from Elsevier. (**B**) Direct competitive immunosensor using AuNPs for the determination of Imidacoprid (IMD), chronoamperometroc measurements and calibration curve. Reprinted from [180], Copyright 2020, with permission from Elsevier.

**Figure 7 biosensors-10-00032-f007:**
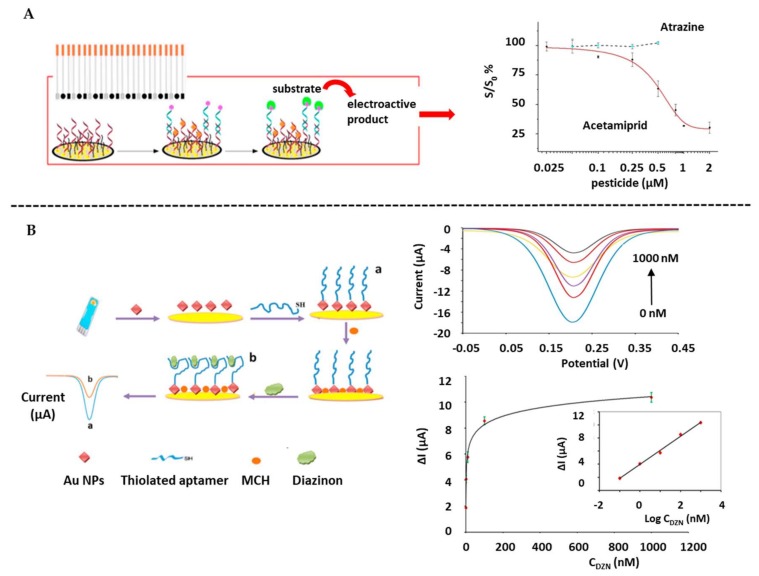
Aptasensors for pesticides determination on screen-printed electrodes: (**A**) Immobilization of aptamer on AuNPs-modified SPGE for the determination of Acetamiprid and the dose-response curve of Acetamiprid and Atrazine. Adapted from [193], Copyright 2016, with permission from Elsevier. (**B**) Immobilization of aptamer on AuNPs-modified gold SPE for Diazinon detection and DPV measurements at different concentrations together with the calibration curve. Reprinted from [195], Copyright 2018, with permission from Elsevier.

**Figure 8 biosensors-10-00032-f008:**
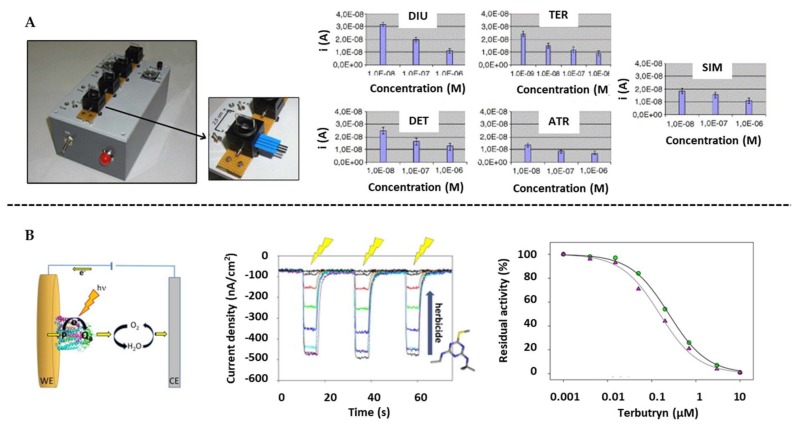
Biological sensors for pesticides determination on screen-printed electrodes: (**A**) Multi-flow detector device for the determination of herbicides using PS II as oxidoreductase. Representation of the current decrease with the herbicide’s concentrations. Reprinted from [199], Copyright 2005, with permission from Elsevier. (**B**) Scheme of a biosensor using bacterial reaction centers on gold SPE for the determination of Terbutryn. Photocurrents detected by LIFT and Terbutryn curves in absence and presence of 80 µM of 2,3-dimethoxy-5-methyl-p-benzoquinone (UQ_0_) (inset the Dixon plot). Reprinted from [200], Copyright 2016, with permission from Elsevier.

**Table 1 biosensors-10-00032-t001:** Enzyme inhibition biosensors reported for pesticides detection on screen-printed electrodes.

Electrode	Enzyme	Analyte	Lineal Range	LoD	Ref
SPGE (array)	AChE	Carbaryl Heptenophos Fenitrothion Dichlorvos Phosphamide	0.8 nM–2.4 µM	0.80 nM 9.2 nM 85 nM 77 nM 130 nM	[74]
	BChE	Carbaryl Heptenophos Fenitrothion Dichlorvos Phosphamide		93 nM 2.8 nM 6.9 nM 14 nM 2390 nM	
Thick-film SPCE	Tyr	Diethyldithio carbamate	–	2 µM	[79]
SPCE	AChE	Dichlorvos Monocrotophs Parathion	16–28 nM	18.1 nM 26.4 nM 14.4 nM	[104]
PB/SPCE	AChE	Aldicarb Carbaryl	63–315 nM 124–497 nM	63 nM 124 nM	[105]
	BChE	Paraoxon Chlorpyrifos-methyl oxon	7–18 nM 1.6–6 nM	7 nM 1.6 nM	
PEDOT:PSS/SPGE	AChE	Chlorpyrifos-oxon	4–760 nM	4.4 nM	[106]
TCNQ/SPGE	AChE	Aldicarb Carbaryl Carbofuran Methomyl	10–500 nM 5–500 nM 1–750 nM 2.5–700 nM	8 nM 4 nM 1 nM 2 nM	[107]
CoPc/SWCNTs/ SPCE	AChE	Paraoxon Malaoxon	18–181 nM 6–159 nM	11 nM 6 nM	[108]
GA/ZnONPs/ SPCE	Tyr	Chlortoluron	1–100 nM	0.47 nM	[109]
Fe_3_O_4_/GR/SPCE	AChE	Chlorpyrifos	0.14–285 nM	0.06 nM	[110]
Al_2_O_3_/SPCE	AChE	Dichlorvos	1–60 µM	0.8 µM	[111]
CB/CoPc/SPCE	BChE	Paraoxon	Up to 100 nM	18 nM	[112]
CoPc/SPCE	AChE	Organophosphates	10^−5^–10^−9^ M	–	[113]
Cyst/GA/AuSPE	AChE	Paraoxon	Up to 145 nM	7.3 nM	[114]
CoPc/CGCE	Tyr	Methyl parathion Diazinon Carbofuran Carbaryl	22.8–379.9 nM 62.4–164.3 nM 22.6–406.8 nM 49.7–248.5 nM	–	[115]
Nf/SPGE	BChE	Trichlorfon Coumaphos	4 × 10^−7^–8 × 10^−7^ M 2 × 10^−7^–5.5 × 10^−6^ M	3.5 × 10^-7^ M 1.5 × 10^-7^ M	[116]
PB/SPGE	ChO	Paraoxon	0.1–1 µM	0.1 µM	[117]
DEP-Au chips	AChE	Paraoxon Carbofuran	–	36.3 nM 36.1 nM	[118]
GA/IrOxNPs/ SPCE	Tyr	Chlorpyrifos	0.01–0.1 µM	3 nM	[119]
SPCE	AChE	Chlorpyrifos	1 × 10^−6^–5 × 10^−2^ M	5 µM	[120]
DCHP/MWCNT/SPCE	AChE	Chlorpyrifos	0.14–2.85 nM	0.14 nM	[121]
Nf/PB/DSPCE	AChE	Isocarbophos Chlorpyrifos Trochlorfon	0.33–16.72 µM	0.33 µM	[122]
SPCE	AChE	Permethrin	6.2–41 µM	8.1 µM	[123]
Cu_3_(PO_4_)_2_/HNFs/ SPCE	AChE/ ChO	Paraoxon	2.18 × 10^−5^–2.18 nM	21.8 fM	[124]
GA/Nf/BSA/ CBNPs/SPGE	BChE	Paraoxon	18.2–109 nM	18.2 nM	[125]
Nf/PB/ZrO_2_/ CNT/SPCE	GMP-AChE	Dimethoate	0.004–43.6 nM	2 pM	[126]
TCNQ/SPCE	BChE	Chlorpyrifos-methyl Coumaphos Carbofuran	3 × 10^−8^–3 × 10^−7^ M 1 × 10^−7^–4 × 10^−6^ M 3 × 10^−8^–1 × 10^−7^ M	20nM 50 nM 10 nM	[127]
PB/SPCE	AChE/ ChO	Chlorpyrifos-methyl Carbofuran	4 × 10^−8^–5 × 10^−7^ M 1 × 10^−8^–1 × 10^−7^ M	30 nM 8 nM	[127]
MWCNTs/SnO_2_/ CHIT/SPCE	AChE	Chlorpyrifos	0.14–2.85 × 10^3^ nM	< 0.14 nM	[128]
CS/PVA NFM/SPCE	AChE	Pirimiphos-methyl oxon	1 × 10^−10^–8 × 10^−9^ M	0.2 nM	[129]
OMC-CHIT/ Fe_3_O_4_-CS/SPCE	AChE	Methamidophos Chlorpyrifos	–	7.09 nM 0.14 nM	[130]
SPSE	AChE	Chlorpyrifos	0–71.3 nM	7.13 nM	[131]
CS/CB/SPCE	AChE	Paraoxon	0.36–1.82 nM	0.18 nM	
MWCNT/SPCE	AChE	Paraoxon	Up to 6.9 nM	0.5 nM	[132]
ZnO/SPCE	AChE	Paraoxon	Up to 5 µM	0.13 µM	[133]
SPGE	AChE	Chlorpyrifos ethyl oxon	0–2 × 10^−8^ M 5 × 10^−8^–2 × 10^−7^ M	3.6 pM	[134]
MWCNT/IL/ SPCE	AChE	Chlorpyrifos	0.14–2.85 × 10^5^ nM	0.14 nM	[135]
PBNCs/rGO/ SPCE	AChE	Monocrotophos	4.5–2688 nM	0.45 nM	[136]
TCNQ/SPGE	AChE	Carbaryl Carbofuran Pirimicard	Up to 5 × 10^−7^ M Up to 1 × 10^−7^ M Up to 5 × 10^−7^ M	10 nM 0.8 nM 0.2 nM	[137]
CoPc/SPCE	AChE	Carbofuran	10^−10^–10^−8^ M	0.5 nM	[138]
TCNQ/Nf/SPGE	AChE	Chlorpyrifos methyl	3–930 nM	68 nM	[139]
TCNQ/BSA/GA/ SPCE	AChE	Paraoxon	1.8 × 10^−7^–5.4 × 10^−5^ M	0.18 µM	[140]
TCNQ/Nf/SPCE	AChE	Carbaryl Parathion methyl	9.9–447.3 nM 3.8–379.9 nM	9.9 nM 3.8 nM	[141]
CoPc/SPCE	AChE	Dichlorvos Parathion Azinphos	1 × 10^−17^–1 × 10^−4^ M 1 × 10^−16^–1 × 10^−4^ M 1 × 10^−16^–1 × 10^−4^ M	fM 0.1 fM 0.1 fM	[142]

SPGE: Screen-printed graphite electrode; PB: Prussian Blue; PEDOT: Poly (3,4-ethylenedioxythiophene); PSS: Poly(styrene sulfonate); TCNQ:7,7’,8,8’-Tetracyanoquinodimethane; GR: Graphene; Cyst: Cysteamine; GA: Glutaraldehyde; AuSPE: Screen-printed gold electrode; CGCE: Acetylcelullose-graphite composite electrode; Nf: Nafion; ChO: Choline oxidase; DEP: Disposable electrochemical printed; DCHP: Dicyclohexyl phthalate; DSPCE: Dual-channel screen-printed carbon electrode; GMP-AChE: Gold magnetic particles-Acetylcholinesterase; HNF: Hybrid nanoflowers; CBNP: Carbon black nanoparticles; CNT: Carbon nanotube; CHIT: Chitosan; PVA: Poly (vinyl alcohol); NFM: nanofibrous membranes; OMC-CHIT: ordered mesoporous carbon–chitosan; SPSE: Screen-printed silver electrode; IL: Ionic liquid; PBNC: Prussian Blue Nanocubes; rGO: reduced Graphene Oxide; BSA: Bovine Serum Albumin.

**Table 2 biosensors-10-00032-t002:** Enzymatic hydrolysis-based biosensors reported for pesticides detection on screen-printed electrodes.

Electrode	Enzyme	Analyte	Lineal Range	LoD	Ref
Nf/SPCE	OPH	Paraoxon Methyl parathion	4.6–46 µM Up to 5 µM	0.9 µM 0.4 µM	[148,149]
MWCNT/SPCE	OPH	Demeton-S	Up to 85 µM	1 µM	[150]
Fe_3_O_4_@Au-NC/ SPCE	MPH	Methyl parathion	1.9–3799 nM	0.38 nM	[151]
SPCE	PH	Parathion	34–343 nM	3.4 nM	[152]
BSA/GA/SPCE	OPH	Diazinon	–	0.59 µM	[153]

Nf: Nafion; OPH: Organophosphorus hydrolase; NC: Nanocomposite; MPH: Methyl parathion hydrolase; PH: Parathion hydrolase; GA: Glutaraldehyde.

**Table 3 biosensors-10-00032-t003:** Direct detection-based biosensors reported for pesticides determination on screen-printed electrodes.

Electrode	Analyte	Lineal range	LoD	Ref
SPCE	Bentazone	0.19–50 µM	34 mM	[157]
CB/SPCE	Carbofuran Isoprocarb Carbaryl Fenobucarb	0.1–100 µM 0.1–100 µM 0.1–100 µM 0.1–100 µM	49 nM 79 nM 48 nM 80 nM	[158]
Thick-film Bi/SPCE	Imidacloprid Thiamethoxam Dinotefuran Clothianidin Nitenpyram	0–110.26 µM	2.97 µM 2.68 µM 7.67 µM 4.12 µM 4.36 µM	[159]
NiO/SPCE	Parathion	0.1–5 µM and 5–30 µM	24 nM	[160]
AuNPs/SPCE	Thiram DEDMTDS Disulfiram	0.29–62.39 µM 0.15–26.62 µM 1.69–50.58 µM	90 nM 50 nM 550 nM	[161]
MIP/AuNPs/ERGO/ SPCE	Cyhexatin	2.60–1298.18 nM	0.52 nM	[162]
CoPc/SWCNT/SPGE	Thiocholine	0.07–0.45 mM	38 µM	[163]
ZnONPs/MWCNTs/ SPCE ZnONPs/Au-SPCE	Glyphosate AMPA	1–10 µM 10–100 µM	300 nM 3 µM	[164]
AG/AuNPs/SPCE	Hydrazine	0.002–936 µM	0.57 nM	[165]
CHIT/ZnO/SPCE	4-nitrophenol	0.5–400.6 µM	230 nM	[166]
Graphene/Nf/SPCE MWCNT/Nf/SPCE	4-nitrophenol	10–620 µM 25–620 µM	600 nM 1.3 µM	[167]
MWCNT-SPE	Sulfentrazone	1–30 µM	150 nM	[168]
AuNP/CeO_2_/SPGE	Hydrazine	0.01–10 mM	–	[169]
CuONPs/SPCE	DCMU	0.5–2.5 µM	47 nM	[170]
NG-PVP/ AuNPs/SPCE	Hydrazine	2–300 µM	70 nM	[171]
Nafion/CNT/SPCE	Paraquat	0.54–4.30 µM	170 nM	[172]
AuNPs/GO/SPCE	Carbofuran	1–30 µM 30–250 µM	220 nM	[173]
Ag@GNRs/SPCE	Methyl parathion	0.005–2780 µM	0.5 nM	[174]
CoHCF/SPGE	Thiocholine	5 × 10^−7^–1 × 10^−5^ M	500 nM	[175]
CB/CoPc/SPCE	Thiocholine	Up to 6 mM	4 µM	[112]

CB: Carbon-black; NP: Nanoparticles; DEDMTDS: N,N-diethyl-N’,N’-dimethylthiuram disulfide; CoPc: Cobalt phthalocyanine; SWCNT: Single-walled carbon nanotube; SPGE: Screen-printed graphite electrode; MWCNT: Multi-walled carbon nanotube; AMPA: Aminomethyl phosphoric acid; MIP: Molecularly imprinting polymer; ERGO: Electrochemical reduction graphene oxide; AG: Activ ated graphite; CHIT: Chitosan; Nf: Nafion; DCMU: 3-(3,4-dichlorophenyl)-1,1-dimethylurea; NG: Nitrogen-doped graphene; PVP: Polyvinylpyrrolidone; GNR: Graphene nanoribbons; CoHCF: Cobalt hexacyanoferrate.

**Table 4 biosensors-10-00032-t004:** Immunosensors reported for pesticides determination on screen-printed electrodes.

Electrode	Analyte	Lineal range	LoD	Ref
Ab/fG-SPCE	Parathion	0.3–3.43 × 10^3^ pM	0.18 pM	[176]
Ab/NH_2_-GQD/SPCE	Parathion	0.03–3.43 × 10^6^ pM	0.16 pM	[177]
PO-SPCE	Chlorsulfuron	0.03–3.88 nM	30 pM	[178]
BSA-IMD/SPCE	Imidacloprid	50–10000 pM	24 pM	[179]
Ab/AuNP/SPCE	Imidacloprid	50–10000 pM	22 pM	[180]
BSA-Ag/Pt/SiO_2_/SPCE	Chlorpyrifos methyl	1.24–62 nM	70 pM	[181]
Ab/PANI/PVS/SPCE	Atrazine	0.02–0.22 µM	4.6 nM	[182]
Ab/ATPh/GA/AuSPE	2,4-D	45 nM–0.45 mM	–	[183]

Ab: Antibody; fG: functionalized graphene; PO: Peroxidase; BSA: Bovine serum albumin; IMD: Imidacloprid; GQD: Graphene Quantum Dots; Ag: Antigen; ATPh: 4-aminothiophenol; GA: Glutaraldehyde; AuSPE: Screen-printed gold electrode; 2,4-D: 2,4-dichlorophenoxyacetic acid.

**Table 5 biosensors-10-00032-t005:** Aptasensors reported for pesticides determination on screen-printed electrodes.

Electrode	Analyte	Lineal range	LoD	Ref
Apt/PANI/AuNPs/SPGE	Acetamiprid	0.25–2 µM	86 nM	[193]
BSA/Apt/rGO-CuNPs/SPCE	Profenofos Phorate Isocarbophos Omethoate	0.01–100 nM 1–1000 nM 0.1–1000 nM 1–500 nM	3 pM 300 pM 30 pM 300 pM	[194]
Apt/MCH/AuNP/AuSPE	Diazinon	0.1–1000 nM	17 pM	[195]

Apt: Aptamer; PAMI: Polyaniline layer; SPGE: Screen-printed graphite electrode; MCH: 6-Mercapto-1-hexanol; AuSPE: Screen-printed gold electrode.

**Table 6 biosensors-10-00032-t006:** Biological sensors reported for pesticides determination on screen-printed electrodes.

Bacteria	Analyte	Lineal range	LoD	Ref
*Spinacia oleracea* *Snecio vulgaris*	DIU ATR SIM TER DET	1 × 10^−8^–1 × 10^−6^ M 1 × 10^−8^–1 × 10^−6^ M 1 × 10^−8^–1 × 10^−6^ M 1 × 10^−9^–1 × 10^−6^ M 1 × 10^−8^–1 × 10^−6^ M	15 nM 13 nM 41 nM 25 nM 24 nM	[199]
*Rhodobacter sphaeroides*	Terbutryn	0.001–10 µM	8 nM	[200]
*Escherichia coli*	Methyl parathion	2–80 µM	0.5 µM	[201]

DIU: Diuron; ATR: Atrazine; SIM: Simazine; TER: Terbuthylazine; DET: Deethylterbuthylazine.

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
