# Peer review of "Electrochemical (Bio)Sensors for Pesticides Detection Using Screen-Printed Electrodes"

_biosensors, 2020, doi:10.3390/bios10040032_

Round 1

Reviewer 1 Report

The manuscript by Pérez-Fernández et al. deals with current applications of electrochemical sensors in pesticide detection using screen printed electrodes. The authors put tremendous work in finding all the relevant literature in this field, which can be seen by over 200 citations in the work. The structure is well suited and the reader can easily follow. Before publication some minor remarks should be addressed by the authors:

  • You may consider adding a small chapter based on used electrochemical techniques in pesticide detection (CV, Impedance spectroscopy, …). As review articles are generally the first point of touch with a new topic, this will help the readers a lot.
  • I would suggest English proofreading, as some passages are grammatically improvable.
  • Line 102 and some other: the citation should be in one parenthesis.
  • Line 228: one blank to much
  • Line 434ff: E. coli should be in italics.

Author Response

Question 1: “You may consider adding a small chapter based on used electrochemical techniques in pesticide detection (CV, Impedance spectroscopy, …). As review articles are generally the first point of touch with a new topic, this will help the readers a lot.”.

Answer: We agree with the referee’s comment. A new section describing the typical electrochemical techniques used for pesticides detection after following one of the (bio)sensing routes schematized in Figure 1 is now included in the revised manuscript (section 2, pages 4-5).

Question 2: “Line 102 and some other: the citation should be in one parenthesis. Line 228: one blank to much. Line 434: E. coli should be in italics.”

Answer: We thank very much the referee for noticing such mistakes, which are now corrected in the revised manuscript.

Reviewer 2 Report

This review has been reviewed for publication at biosensors. In this work authors presented a review on using screen printed electrodes for the detection of pesticides using electrochemical (bio)sensors. The review has a good quality and I cannot find any scientific problem in it. I recommend for acceptance.

Author Response

General comment: “This review has been reviewed for publication at biosensors. In this work authors presented a review on using screen printed electrodes for the detection of pesticides using electrochemical (bio)sensors. The review has a good quality and I cannot find any scientific problem in it. I recommend for acceptance”.

Answer: We thank very much the referee for his/her positive feedback.

Reviewer 3 Report

The manuscript « Electrochemical (bio)sensors for pesticides detection using screen-printed electrodes” by Pérez-Fernández et al., describes use of SPEs in the development of biosensors for detection of pesticides in food and water.

The paper can be published in Biosensors after minor modifications.

  1. The inhibition of the reaction catalyzed by AChE is used to determinate OP concentration. Different works obtained detection of OP with quite different LODs. Please comment why in some cases LOD was improved. In the present form, the authors only listed LODs (line 115-128). It looks like all given papers have the same sensor but obtained various efficiency.
  2. Figure 2 should be completed. As it is provided in Introduction part, a schema explaining possible surface functionalization should be added to Figure 2, like nanoparticle grafting (by adsorption?), aptamer attachment (by thiol-Au covalent link), antibody immobilization (by…). In this way, the possibilities to employ SPEs will be provided.
  3. Resolution of all Figures should be improved.
  4. Conclusion is too repetitive comparing the whole manuscript. I propose that authors compare presented papers and propose the best strategy for the moment regarding sensitivity, stability, and selectivity. Is any sensor available commercially?
  5. The sentence in the Abstract :

“in this review article we summarize and give an authoritative opinion about the use of SPEs as analytical tools in the development of (bio)sensors for pesticides of interest for food control.

Should be modified, because no authoritative opinion about the use of SPEs is provided.

Author Response

Question 1: “The inhibition of the reaction catalyzed by AChE is used to determinate OP concentration. Different works obtained detection of OP with quite different LODs. Please comment why in some cases LOD was improved. In the present form, the authors only listed LODs (line 115-128). It looks like all given papers have the same sensor but obtained various efficiency”.

Answer: We thank the referee for bringing up to this important point. The different affinity between each analyte and the enzyme is the key factor affecting the sensitivity/LOD of the assay. A short explanation clarifying this point is now included in the revised section 3.1 (page 6)

 Question 2: “Figure 2 should be completed. As it is provided in Introduction part, a scheme explaining possible surface functionalization should be added to Figure 2, like nanoparticle grafting (by adsorption?), aptamer attachment (by thiol-Au covalent link), antibody immobilization (by…). In this way, the possibilities to employ SPEs will be provided”.

Answer: We agree with the referee’s comments. The main methods used for the immobilization of (bio)receptors on the working electrode of SPEs for pesticides detection are now illustrated at the new Figure 2B (page 3) in the revised manuscript.

Question 3: “Resolution of all Figures should be improved”.

Answer: We thank the referee for his/her comment. Figures are provided in the maximum resolution given by the original sources. Probably the problem comes from the pdf conversion. Anyway, a word file containing high quality figures has been sent to the editorial office.

Question 4: “Conclusion is too repetitive comparing the whole manuscript. I propose that authors compare presented papers and propose the best strategy for the moment regarding sensitivity, stability, and selectivity. Is any sensor available commercially?”.

Answer: We thank the referee for his/her comments. Actually, in the conclusions section we state that the (bio)sensing strategy to be followed must be carefully studied for each concrete case. The selectivity and sensitivity/detection limit required and the availability of specific enzymes and antibodies are the main parameters that should define the detection strategy more suitable for each particular application.

As far as we now, none of the reported electrochemical (bio)sensors for pesticides detection on screen-printed electrodes is commercially available yet. Efficiency and long-term stability of the enzymes and the antibodies are probably the key issues limiting their commercial implantation. Moreover, in the case of the immunosensors, the most challenging point is related to the sampling, washing, etc. steps required, which limits their use by non-skilled people. The combination with microfluidics seems to be of key relevance here, so high efforts in this sense are previewed for the coming years.

All these issues are now better stated in the revised conclusions section of the manuscript.

Question 5: “The sentence in the Abstract: “in this review article we summarize and give an authoritative opinion about the use of SPEs as analytical tools in the development of (bio)sensors for pesticides of interest for food control”. Should be modified, because no authoritative opinion about the use of SPEs is provided.”.

 Answer: In agreement with the referee’s comment, last sentences of the abstract section have been modified in the revised manuscript as follows: “In this context, in this review article we summarize and discuss about the use of SPEs as analytical tools in the development of (bio)sensors for pesticides of interest for food control. Finally, aspects related to the analytical performance of the developed (bio)sensors together with prospects for future improvements are discussed”.